# Electrodeformation of White Blood Cells Enriched with Gold Nanoparticles

**Nicholas G. Hallfors** [1], **Jeremy C. M. Teo** [2], **Peter M. Bertone** [3,4], **Chakra P. Joshi** [5], **Ajymurat Orozaliev** [2], **Matthew N. Martin** [5] and **A. F. Isakovic** [4,*]

1   Biomedical Engineering Department, Khalifa University, Abu Dhabi P.O. Box 127788, United Arab Emirates; nicholas.hallfors@ku.ac.ae
2   Mechanical and Biomedical Engineering Department, New York University Abu Dhabi, Abu Dhabi P.O. Box 129188, United Arab Emirates; jeremy.teo@nyu.edu (J.C.M.T.); ajymurat.orozaliev@nyu.edu (A.O.)
3   Department of Bioengineering, University of Pennsylvania, Philadelphia, PA 19104, USA; pbertone@seas.upenn.edu
4   Department of Physics and Astronomy, Colgate University, Hamilton, NY 13346, USA
5   Physics Department, Khalifa University, Abu Dhabi, P.O. Box 127788, United Arab Emirates; chakraprasadjoshi@yahoo.com (C.P.J.); matthew.martin@ku.ac.ae (M.N.M.)
*   Correspondence: aisakovic@colgate.edu or iregx137@gmail.com

**Abstract:** The elasticity of white blood cells (WBCs) provides valuable insight into the condition of the cells themselves, the presence of some diseases, as well as immune system activity. In this work, we describe a novel process of refined control of WBCs' elasticity through a combined use of gold nanoparticles (AuNPs) and the microelectrode array device. The capture and controlled deformation of gold nanoparticles enriched white blood cells in vitro are demonstrated and quantified. Gold nanoparticles enhance the effect of electrically induced deformation and make the DEP-related processes more controllable.

**Keywords:** leukocytes; electrodeformation; lab-on-chip; nanoparticles; Young's modulus

## 1. Introduction

Cells are subjected to a variety of mechanical forces in vivo, and the way they deform in response to mechanical, electrical, and biochemical stimuli relies on a combination of passive and active processes [1]. Red blood cells significantly deform as they travel throughout the body's capillary networks, which are at times smaller than the cells' resting size. Diseases such as malaria and sickle cell anemia are associated with disruption of the cell membrane elasticity, leading to capillary blockages and a loss of oxygenation [2]. The deformability of cells has even been linked to cancer, where highly metastatic cells have been shown to be soft and deformable, allowing them to migrate through tissue into the blood stream [3–5].

In white blood cells (WBC), quantification of a cell's elastic modulus via deformability measurements could provide insights into the physiological state of the cell. HL60 cells can differentiate into monocytes, granulocytes, or macrophages, and mechanical deformation alone can distinguish which pathway the HL60 cell will take [6]. Neutrophil activation leads to reduced deformability, which has been demonstrated by morpho-rheological (MORE) analysis [7]. Monocytes from individuals afflicted by Respiratory Tract Infection (RTI) or Acute Lung Injury (ALI) both increased in size with staphylococcus stimulation, but only viral RTI monocytes displayed any measurable increase in deformation. These results indicate that size and deformation studies may be able to identify the presence of viral, bacterial, or other inflammatory diseases through lymphocyte mechanical analysis, implying a need for fast, reliable methods to measure mechanical properties.

A number of methods currently exist to deform and measure the elasticity of cells. Direct methods such as AFM and parallel plate rheometry [3,8,9] involve physically deforming a cell by applying force with a contact probe and measuring probe deflection. Optical tweezers and optical stretchers utilize laser light to deform cells and measure elastic modulus [5,10–12]. Optical deformation has even been used to demonstrate the identification of noncancerous, cancerous, and metastatic cells in a mixed population [13].

Dielectrophoresis (DEP) is a method of manipulating particles suspended in a fluid medium, whereby a particle placed in a non-uniform electric field experiences a force through its dielectric response. The amplitude of the DEP forces experienced by the cells is modulated by the dielectric properties of the cell and surrounding media and is expressed by the Clausius–Mossotti (CM) factor, which will vary from -1 for a strongly repellent force, to +1 for a strongly attractive force [14]. Under certain conditions, a cell may become trapped and deformed by DEP, a phenomenon hereby referred to as electrodeformation.

A number of studies have utilized electrodeformation to controllably deform cells to measure mechanical characteristics. Electrodeformation of red blood cells (RBCs) have been performed with various devices utilizing DEP [15–17] and references therein, as well as more sophisticated trap and release microarrays for high-throughput imaging and characterization [18]. The majority of such studies have involved red blood cells [19] due to their readily available nature and easily observable electrodeformation behavior that follows the analytical prediction from the model to be discussed hereafter. We wish to emphasize that the overall review of the electrodeformation of RBCs is beyond the scope of this manuscript.

In contrast with RBCs and despite a number of promising initial studies, a fast, reliable, highly parallelized, and scalable method for the controlled deformation and observation of white blood cells (WBCs) is yet to be realized. In the current work, a microelectrode array was designed to accommodate several cells at once for rapid, parallel capture, deformation, and imaging (Figures 1 and 2 here) of WBCs. An additional specific contribution of this report is the use of gold nanoparticles (AuNP) introduced to the cells to enhance the effect of electrodeformation.

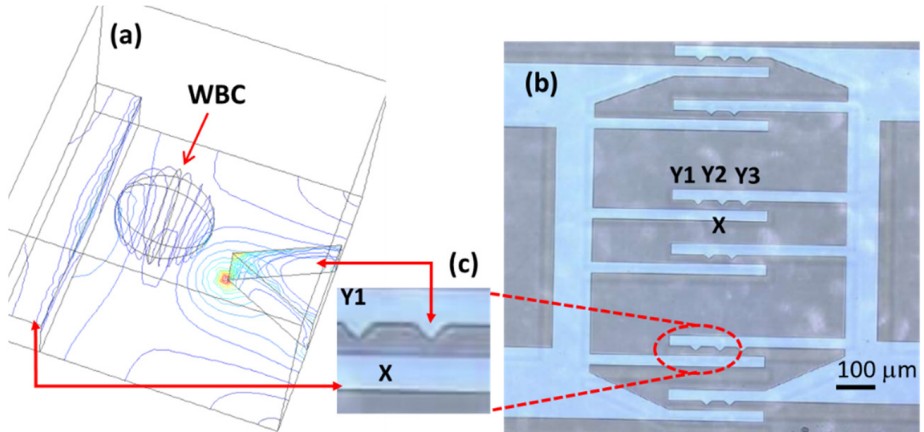

**Figure 1.** (**a**) 3D numerical simulation of electrodes and cell (electrode height not to scale) showing electrical potential lines in a representative medium. (**b**) Optical microscope image of ITO electrodes on glass, with electrodes labeled (X, Y1, Y2, Y3). (**c**) Closeup of a fraction of electrode array at the region of highest field strength. Cells are attracted to this region by DEP forces and fixed at electrode tips, where they are subsequently deformed.

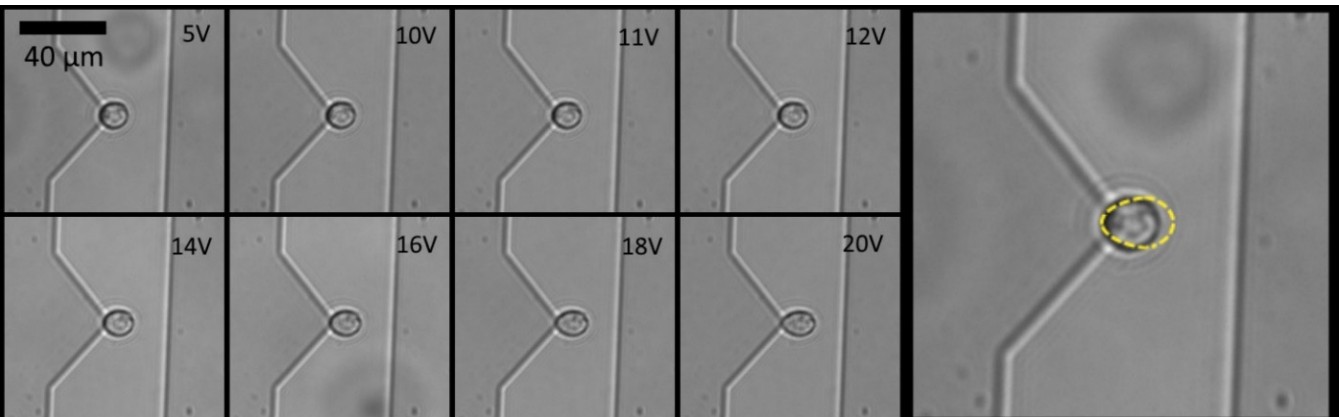

**Figure 2.** THP-1 cell captured at electrode tip by DEP forces and subsequently deformed. Right side shows an overlay of the 20V deformation on top of the resting cell. Supplementary Materials contain the movie file of this process.

## 2. Materials and Methods

### 2.1. Dielectrophoretic Electrodes on a Chip

Standard microfabrication techniques were used to produce indium tin oxide metal electrodes on a glass substrate. Indium tin oxide (ITO), 150 nm thick, was deposited by electron beam evaporation on a borosilicate glass wafer. A lift-off process was used to pattern electrodes on the substrate. The patterned electrodes were subjected to an annealing process at 400 deg C in air. The high-temperature annealing of ITO while increasing oxidation results in an ITO film that is highly transparent and conductive (Figure 1) [20–22]. Following lift off, glass wafers were diced with a dicing saw. One electrode is a simple rectangular prism (labeled "X"), while the other one has a "saw tooth" pattern where each tip ("Y1" etc.) captures one cell at the time. Additional details are available in Supplementary Materials. The geometrical details of the electrodes' lithographic pattern were obtained using numerical modeling of the influence of various electrodes' geometries and dimensions with the goal of having considerable induced dipole moment at the cell, but without strong electric field gradients that would increase a probability of cells being damaged.

### 2.2. Cells

Human THP-1 and Jurkat cell lines (ATCC) were grown in RPMI 1640 medium containing D-glucose, HEPES, L-glutamine, sodium bicarbonate, and sodium pyruvate (Gibco) supplemented with 10% fetal bovine serum (FBS, Gibco) and 1% penicillin–streptomycin (Biosera). For cells with gold nanoparticles, cells were incubated in media with a small volume of added AuNP colloid for 24 h [23,24]. Uptake of AuNP was verified visually and by X-ray fluorescence (XRF) (Figure 3, here).

### 2.3. Gold Nanoparticles (AuNP)

Synthesis of naked AuNPs was performed following the method of Martin et al. [25]. AuNPs are generally considered biocompatible and are being investigated for their applications in medicine and research as carriers for bioactive compounds [23], as contrast agents [26], or as radiation absorbent materials [27]. A 40 mL capacity clean borosilicate glass vial containing 9.25 g of de-ionized water was mixed with an aqueous gold precursor solution 0.1 g (~100 μL) containing 50 mM $HAuCl_4$/HCl producing a light yellow solution. To this light yellow solution, 0.65 g (~650 μL) of freshly prepared aqueous 50 mM $NaBH_4$/NaOH was added rapidly while vortexing. Upon the addition of the alkaline borohydride solution, the reaction mixture immediately turned red, signaling the nucleation of gold nanoparticles at room temperature, and was vortexed for one minute. The ruby red AuNP solution was then placed in a metal heating block (already at 250

°C) for three minutes to grow the AuNPs and improve the monodispersity. The vial was then quenched for 30 s under running water to arrest the kinetic growth of NPs. These hydroxide-stabilized naked AuNPs were then used for the WBC experiments here. More details are in Supplementary Materials.

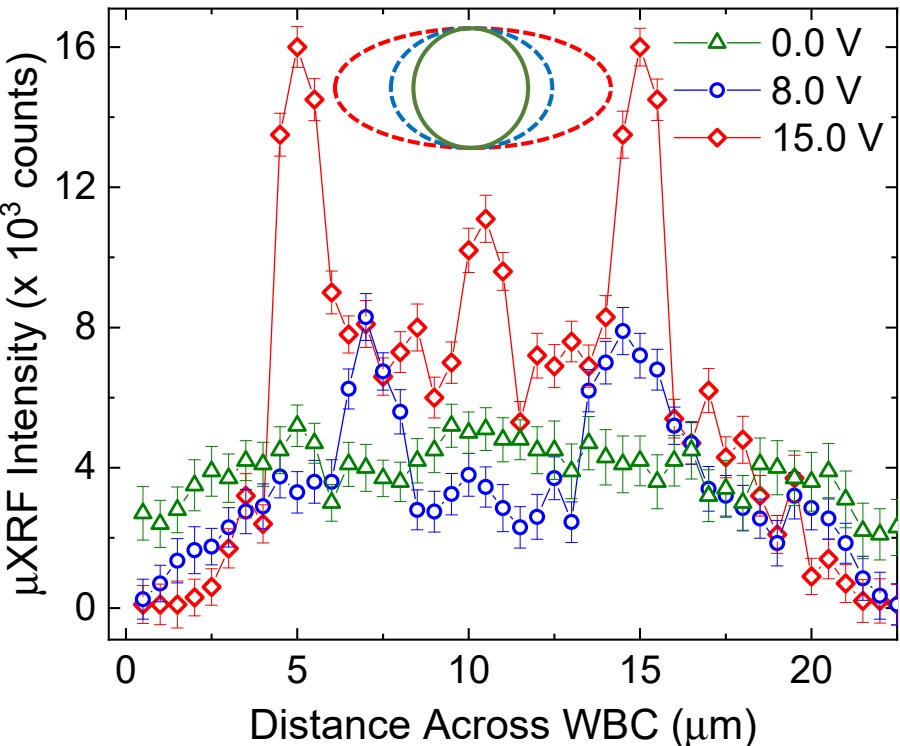

**Figure 3.** Micro-X-ray fluorescence scans performed while cell and gold nanoparticles are under the influence of an applied electric field. Zero voltage shows relatively evenly distributed AuNPs. As the voltage is increased, the cell is deformed, and the concentration of AuNPs in the membrane region is increased. The upper centered insert shows sketched deformation of the WBC (not to scale), under the assumption that the deformation is elliptical.

### 2.4. Apparatus

Electrode array chips were placed on glass slides and connected to a Rigol DG4062 function generator via copper tape and coaxial cable. Cells were centrifuged at 1000 rpm for 5 min, and RPMI was aspirated. A DEP media composed of 8.5% (*w/v*) sucrose and 0.3% (*w/v*) dextrose in distilled water was adjusted to 10 mS/m by adding a small volume of phosphate buffered saline (PBS). Cells were resuspended in the DEP media at low density. A small volume of cell suspension was placed on the electrodes, and AC current was applied. For deformation experiments, frequency was fixed at 700 kHz, while voltage was stepped from 0–20 Vpp in 1V increments. Cells were imaged on a Zeiss Observer Z1 microscope in bright field mode.

DEP forces induced by the field asymmetry pulled cells toward the electrode tips, where the field was highest (Figure 2). Once at the tip, cells became fixed in place and began to deform as voltage was increased (Figure 2). Voltages above 20.0 V generally caused cells to span the electrode gap, doing permanent damage to the cell, so voltage was limited to 20.0 V for the purpose of this study. The electrically induced deformation is reversable and repeatable for at least 10 cycles under the conditions outlined in this Report.

### 2.5. Image Analysis

Images were imported to ImageJ, and cell dimensions were manually measured, frame by frame. An elliptical selector was used to best fit the cell during deformation, and all cells are assumed to be elliptical at all time points. Ellipse geometries were then exported for analysis.

## 3. Results

### 3.1. Electrode Array Fabrication

Indium tin oxide on glass was chosen for electrode fabrication due to ideal optical and electrical properties. The electrode geometry was designed in such a way as to accommodate single cells, positioned by DEP forces prior to deformation (Figure 1). The small gap size allows us to keep voltage low, while the electrode spacing was chosen to balance the need for separation between cells with parallelization of the process. One could, in principle, expect some variations of the spatial 3D profile of the electric field and the equipotential lines by varying the size, the shape, and the spatial distribution of the electrodes. The electrode design shown here is a result of numerically optimized design, but we do not exclude the possibility that additional experimental work with the electrodes' influence on WBCs could lead to insights not reported here. Additional details of the microfabrication process are available in the Supplementary Materials file.

While a more thorough study of cell viability would have been helpful, we report that all cell data in this report have been viable up to 20 volts. Above this voltage (often as low as 22 volts and higher), WBCs become either permanently deformed or stuck at the tip of an electrode (Y1,2,3 spots in Figure 1b,c), or both.

### 3.2. X-ray Microfluorescence for Detection of Gold Nanoparticles Absorption

Our early tests demonstrated a promising response of white blood cells to DEP forces in the presence of gold nanoparticles (AuNPs). This has motivated the desire to establish where exactly are AuNPs located under the experimental conditions in this report.

To this end, electrodeformation experiments were conducted while the electrode array device and WBCs were under X-ray illumination. Figure 3 here shows the result of changing spatial concentration of AuNPs while scanning with a sub-micron resolution (nominal resolution 250 nm $\pm$ 75 nm) is performed. Prior to the application of the electric field, the spatial distribution of AuNPs is approximately even, as there is no region inside or outside the WBCs that generates a stronger fluorescent signal. The situation changes appreciably after the electric field is turned on, so at 8.0 V we see the peaks, the separation of which roughly approximates the width of the white blood cells studied. It is also possible to detect some decrease in the fluorescence signal outside the cell and at the cell's center. Further increase in the applied electric field produces additional peaks in the fluorescence distribution, indicating that a majority of AuNPs become adsorbed at the cell membrane (either inside or outside the cell), with some additional concentration in the area of the cell's center. We note that there are uncertainties in these data originating in the averaging of the XRF signal over several cells and in the positioning. The uncertainties for the horizontal axis (distance across) measurement are omitted, as they are well approximated by the size of the symbol. This approach could be seen as a pilot study leading into additional insights about function of the cells when enriched with AuNPs [28], but we emphasize the result presented in Figure 3 serves only to confirm that there is a change in the spatial distribution of AuNPs in and around the WBCs as one changes the applied electric field.

### 3.3. Frequency-Dependent Response

DEP forces exerted upon a cell are dependent upon frequency, which is often reflected in the behavior of the CM factor. The CM factor profile is different for each cell type, so

understanding the CM profile is an important piece of any DEP study. A study of the frequency response was performed on both THP-1 and Jurkat cells. This was accomplished by keeping the voltage constant at 10 V while varying the signal frequency and measuring the time needed for nearby cells to reach the electrode tip. This time is inversely related to the attractive force and serves as a good initial approximation of the CM factor [29–31]. Due to the design of our device, negative DEP forces are very small, and this method fails as the frequency approaches the cell's crossover value and time goes to "infinity".

Extrapolation is used to predict the cell's crossover value based on the measurements approaching negative DEP (Figure 4 (L)). Specifically, polynomial fitting and extrapolation lead us to values for crossover frequencies $f_{xo1,Jurkat}$ = 21.5 kHz $\pm$ 3.4 kHz and $f_{xo1,THP\text{-}1}$ = 57.7 kHz $\pm$ 3.4 kHz. These values are broadly in agreement with a number of related studies of white blood cells [32–36]. We wish to emphasize that the main purpose of this report is the deformability study, so this measurement has been conducted primarily as an overall quality check, and its focus was not to suggest significantly different values of crossover frequencies for Jurkat and THP-1 white blood cells. These experimental results motivated our need for better understanding the effect of introducing AuNPs into WBCs on electrostatic properties of WBCs. With this in mind, we modeled dielectric responses of WBCs using Claussius–Mossotti's approach in a double-shell approximation (See Supporting Materials).

We have used the output of this model and the prescription offered by [37] to generate the plot in Figure 4 (R), for the imaginary component of complex permittivity. The white blood cell parameters are available (references here and Supporting Materials), and approximating the influence of AuNPs as a change of the relativity permittivity of the medium WBCs is used for this experiment. Understandably, this approximation is a simplification, but we suggest that, to a first order, scalar modification of permittivity is one way to understand the role of AuNPs. Details of the model are provided in Supporting Materials, specifically in part SOM-6.

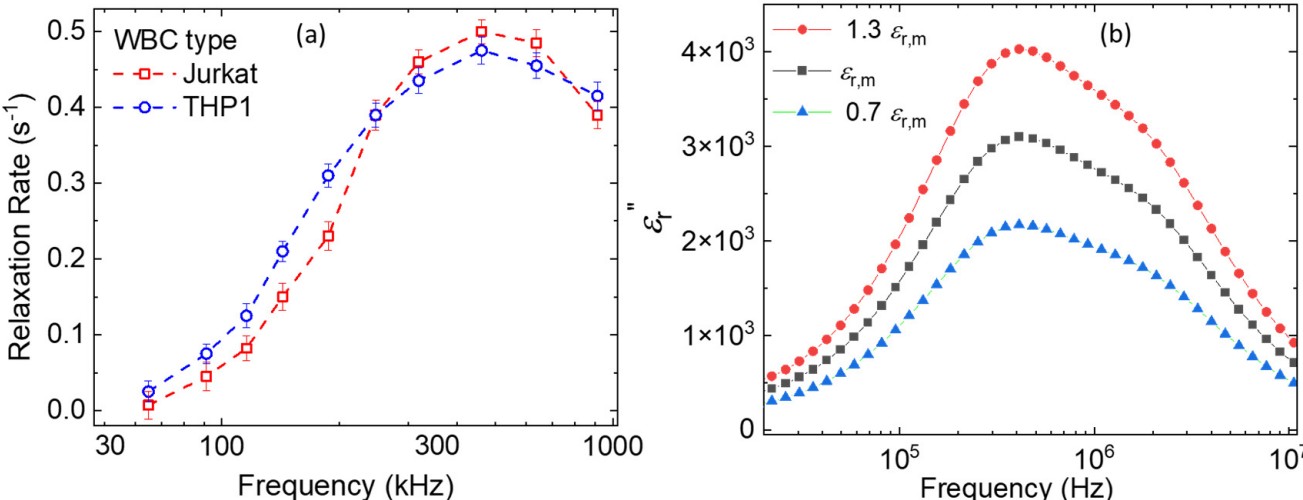

**Figure 4.** (**a**) The inverse of the time of the "attraction until rest" is recorded at different frequencies and the fixed applied electric field. In the text we elaborate how these data are (1) fit to a polynomial and then (2) extrapolated to obtain the reported crossover frequencies. (**b**) Claussius–Mossotti's multi-parameter model indicates a possible range of changes of the relative impedance upon addition of Au NPs that allows WBCs to achieve the peak at a similar frequency observed in (**L**).

The results in Figure 4 have, in part, informed our motivation to perform the deformation study at 700 kHz, which is lower than the MHz range where the relaxation rate and the relative permittivity both start dropping but more than an order of magnitude above the range where the crossover frequencies occur. Additional studies are necessary

to understand the response of WBCs to a broader range of frequencies, especially in the overall region [100 kHz, 3 MHz].

### 3.4. Deformation Study

Two cell lines were studied: THP-1 monocytes and Jurkat's T-lymphocytes. The cells were deformed without gold nanoparticles (AuNPs) modification, then after incubation with plain AuNPs, and AuNPs modified with PEG and citrate. Cells were suspended in isotonic, nonconducting media for DEP. DEP forces attracted single cells to the electrode tip, where they were trapped and subsequently deformed. A gradual deformation was induced by increasing voltage in 1.0 volt increments at 700 kHz. Very little deformation was seen until the voltage reached a value of 5.0 volts for most cells, after which cells stretched into a shape well approximated by an ellipsoid (ellipse in 2D), with the aspect ratio of the ellipse increasing as the applied voltage grows. Voltage was limited to a 20 V maximum, as beyond this point cells suffered permanent damage. This is not an impediment to scale-up of this method, as we will see that the voltages below 20.0 V are sufficient for controllable deformation and cell sorting protocols. If cell density is too high, the likelihood of multiple cells chaining together to span the electrode gap increases. In this case, the cells do not fully deform but become permanently trapped at the electrodes, so we have avoided performing measurements for more than one cell. It is not impossible to rely on multi-cell capture, but the findings from such work require more detailed statistical analysis, and this report focuses on simple, easy-to-repeat experiments.

Figure 5 shows raw deformation data for both types of cells and for variations in the utilization of AuNPs (PEG, citrate). We have tested at least 10 cells, one cell at a time, for each of the sample variations reported. We note that, as of yet, this is not a high-throughput process. Some observations that follow from this are:

(a) It is apparent that very little deformation occurs below 5.0 V.
(b) It takes some additional voltage increase (+2 to 3 volts, in general) to start observing statistically significant differences for various modifications of WBC with AuNPs.
(c) As expected, the larger the applied voltage, the more difference between deformation curves is observed
(d) The deformation curves (the stretch ratio vs the applied voltage) while generally non-linear, are not as apparently quadratic in voltage as one would expect from numerous electro-deformability studies conducted for red blood cells.

To better understand data in Figure 5, and ultimately to check for the predicted quadratic dependence [14,27] of the stretch ratio on applied voltage, we have performed the fit of the stretch ratio as a linear function of the square of the applied voltage. From the deformation data, we were able to calculate the shear elastic modulus for the cells by a Maxwell force model modified from Engelhardt and Sackmann [17].

$$F_{\text{DEP}} = 2\pi r^3 \varepsilon_m Re\{K(\omega)\}\nabla|E_{rms}|^2 \tag{1}$$

where $r$ is the particle radius, $\varepsilon_m$ is the absolute permitivity of the fluid, and

$$K(\omega) = \frac{\varepsilon_p^* - \varepsilon_m^*}{\varepsilon_p^* + 2\varepsilon_m^*} \tag{2}$$

is the Clausius–Mossotti factor, which describes the polarizability of the particle. $E_{rms}$ is the root-mean-square amplitude of the electric field. These calculations rely on the Maxwell stress, which predicts a dependence of the deformation as the square of the voltage.

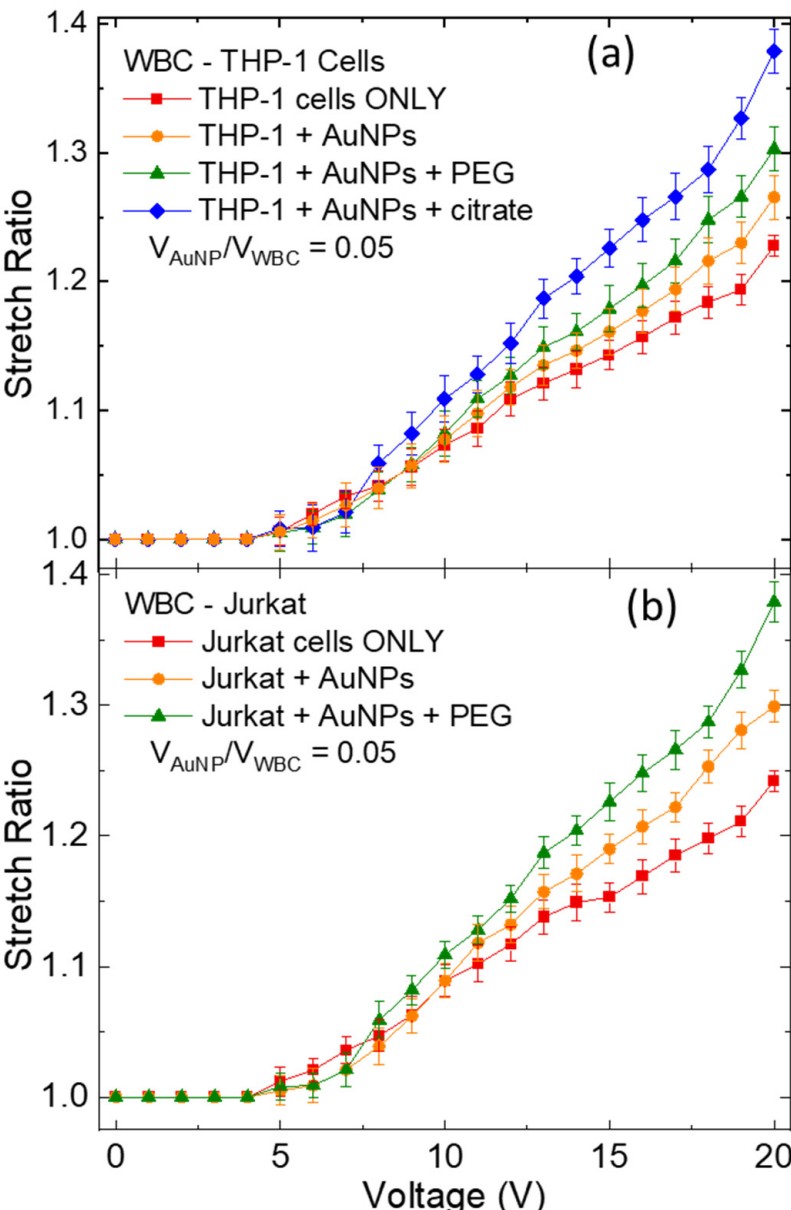

**Figure 5.** (**a**) Stretch ratio measured as a function of the applied voltage for THP-1 white blood cells (WBC). In addition to "cells only" and "nanoparticles enriched WBCs", we used PEG-ylated and citrated solutions of AuNPs, as discussed in the main text. (**b**) Same study, but for the Jurkat type WBCs. Elliptical deformation is assumed during analysis of cells' images. The concentration of AuNPs, in the form of a volume fraction of the overall cell medium, was kept at 5%.

As can be seen from Figure 6 here, the data demonstrate an overall good fit to $V^2$ ($\kappa_R^2$ in the range 0.95 and 0.98 for all fitting shown), but with a caveat that there is a region of data overperforming and underperforming the parabolic fit. One way to interpret data in Figures 5 and 6 here is to propose the existence of different regimes of electro-deformability for white blood cells. For example, based on the intersection points between the data and the straight fit lines in Figure 6, it seems that these regimes would be approximately (i) 0.0 V–8.0 V, (ii) 8.0 V–15.0 V, and (iii) 15.0 V–20.0 V. Assuming this empirically driven qualitative argument has a basis in the fundamental response of the {WBCs + AuNPs} system, it would indicate that a more detailed, molecular level model for the behavior of the {WBCs + AuNPs} system in an electric field is called for. Development of such model is beyond the scope of this report, however. It would likely start by accounting for (i) an

increase in dielectric constant due to the insertion of AuNPs and/or (ii) a charge separation of the counterions from the double layer occurring before migration of AuNPs.

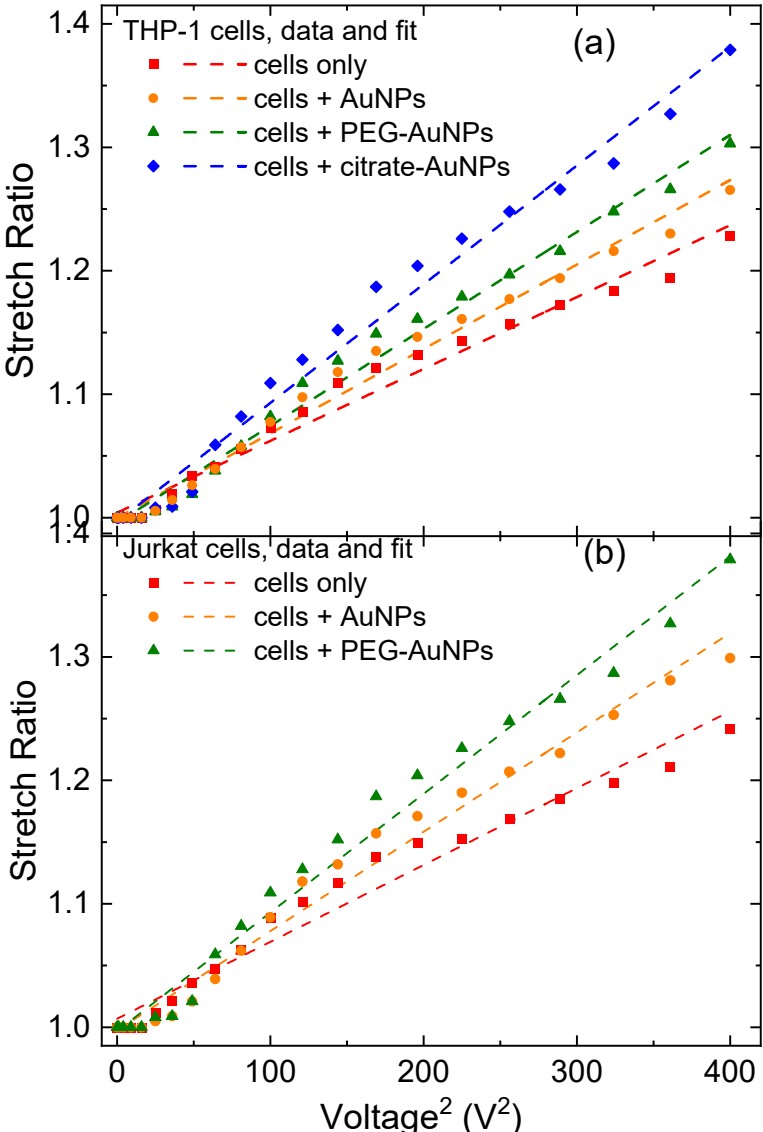

**Figure 6.** (**a**) Stretch ratio plotted as a function of the squared applied voltage, for THP-1 white blood cells (WBC), for the purpose of linear fit. Points represent the same values as in Figure 5, while the dashed line represents the fit values. It is clear that there are different deformation regimes, as indicated by the crossover points (where the imaginary line connecting the points crosses the line of the fit); (**b**) same analysis for Jurkat cells.

Finally, based on electro-deformability data from Figure 5 and fitting analysis in Figure 6, we have determined the elastic modulus for the types of WBC we studied here. Results reported in Figure 7 show a significant reduction in the effective elastic modulus for both THP-1 and Jurkat cells with the addition of AuNP, and further reduction with nanoparticle modification. Some possible mechanisms for AuNP effected modulus changes are as follows:

(1) Particles embedded within the cell membrane or floating within the cytoplasm change the electrical impedance enough to affect the permittivity of the system.
(2) Nanoparticles provide additional active surface area for electrical forces, effectively increasing the surface area of the cell without increasing its resting radius.

(3)   Furthermore, AuNP modified with molecular conjugates PEG and citrate showed additional changes to the effective elastic modulus for WBCs.

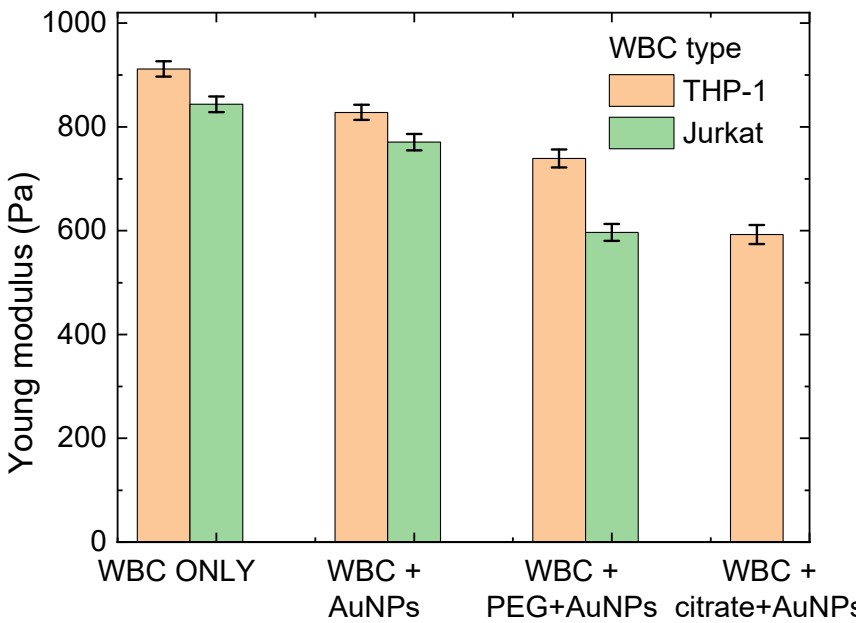

**Figure 7.** Values of the Young modulus for different experimental cases in this report.

### 4. Conclusions

We have chosen dielectrophoresis (DEP) as a method to examine the Young modulus of white blood cells (WBCs) because it offers more than one externally controlled design parameter (voltage, frequency, shape of electric field lines due to control of the geometry of the electrodes), and we enriched WBCs with AuNPs leading to more controllable WBCs.

We have fabricated a novel microelectrode array for the controlled deformation of WBCs. THP-1 monocytes and Jurkat's T-lymphocytes were enriched with gold nanoparticles; were deformed with electrostatic forces; and based on the geometry of the device and applied voltage and cell size, Maxwell tension and cell elastic modulus were determined.

This method was developed because despite the existence of several methods for measuring white blood cells' elastic moduli, they often give different results for similar cell types. Introduction of AuNPs helps in this regard because they modify the dielectric constant of WBCs, allowing for a tunable control of the modified cells' electrical dipole moment, via a frequency-dependent applied electric field. In addition to the cell elastic modulus, we report on spatial resolution of the adsorbed AuNPs and on crossover frequencies in the standard Claussius–Mossotti model of response.

The nature of the electrodeformation data indicates that novel efforts are needed to better understand the response on white blood cells to electric fields on a molecular level. The device we have developed is likely to lead to a highly parallel measurement of cells with a high degree of confidence for high-throughput characterization. Overall, these results represent a useful addition to the rapidly growing field of on-chip quantifiable cell properties [38–51]. It can be expected that improvements in the approach here bring additional progress towards high-throughput cell quantification methods, such as those reported in [52–54], including confocal laser scanning microscopy [52], fluorescence lifetime imaging [53], and third harmonic generation cytometry [54].

**Supplementary Materials:** The following are available online at https://www.mdpi.com/article/10.3390/pr10010134/s1, Figure S1: A 3D model of electric field lines for two electrodes, top and side view, Figure S2: A side of the main steps of the microfabrication, Figure S3: A movie depicting the deformation of a single white blood cell with gold nanoparticles, as the voltage increases; Figure S4:

An electron microscope image of a 2D array of AuNPs and a study of their size; Figure S5: UV-VIS spectra of some of the AuNPs with varied ligands; Figure S6: Some additional details of modeling.

**Author Contributions:** Conceptualization, J.C.M.T. and A.F.I.; device fabrication, N.G.H., A.O. and A.F.I.; measurements, N.G.H. and A.F.I.; formal analysis, N.G.H., P.M.B. and A.F.I.; Gold nanoparticles production, M.N.M., C.P.J. and A.F.I.; writing—original draft preparation, N.G.H., A.F.I. and P.M.B.; writing—review and editing, N.G.H., J.C.M.T., P.M.B., C.P.J., A.O., M.N.M. and A.F.I. All authors have read and agreed to the published version of the manuscript.

**Funding:** The research was funded internally at KUST through KUST Internal Research funds.

**Institutional Review Board Statement:** Not applicable.

**Informed Consent Statement:** Not applicable.

**Data Availability Statement:** The data presented in this study are available upon reasonable request.

**Acknowledgments:** The authors thank KU BME Dept. for the use of equipment, KU—MASDAR Institute for the access to Microfabrication Facility, and NYU-Abu Dhabi for the assistance in mask making. We express our gratitude to B. Samara and L. George (KU) for technical assistance. We are especially thankful to our colleagues at Brookhaven National Lab for the access to X-ray fluorescence setup and for providing additional cells and chemicals. BNL-CFN and BNL-NSLS-II are funded by the US Department of Energy. NGH acknowledges Graduate Scholarship Fund and HEIC at KU. J.T., M.N.M. and A.F.I. acknowledge KU Internal Research Funds. A.F.I. acknowledges hospitality and support at Cornell CNF, supported by US NSF.

**Conflicts of Interest:** The authors declare no conflict of interest.

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
