# Peer review of "Electrodeformation of White Blood Cells Enriched with Gold Nanoparticles"

_processes, doi:10.3390/pr10010134_

Round 1
Reviewer 1 Report
See the attached document.

Reviewer 2 Report
1, In the introduction part, some methods were listed to show deformation and measurement of cell elasticity. Compared with the other methods, could you please add the advantages of DEP method?
2, In the introduction part, compared with RBCs, is there any other high throughput imaging and characterization method for WBCs?
3, Do the tip shape and gap size will affect the deformation of WBCs? If so, what do you expect? If not, why? This discussion can be added to the manuscript if possible.
Round 2
Reviewer 1 Report
Thanks for the authors' efforts to revise the manuscript. I believe the quality of this paper has been improved.